# A Noise Reduction Method for Four-Mass Vibration MEMS Gyroscope Based on ILMD and PTTFPF

**DOI:** 10.3390/mi13111807

**Published:** 2022-10-23

**Authors:** Zhong Li, Yikuan Gu, Jian Yang, Huiliang Cao, Guodong Wang

**Affiliations:** 1Shanxi Software Engineering Technology Research Center, Taiyuan 030051, China; 2School of Software, North University of China, Taiyuan 030051, China; 3Key Laboratory of Instrumentation Science & Dynamic Measurement, Ministry of Education, North University of China, Taiyuan 030051, China; 4Beijing Institute of Aerospace Control Devices, Beijing 100039, China

**Keywords:** four-mass vibration MEMS gyroscope, local mean decomposition, sample entropy, parabolic tracking time-frequency peak filtering

## Abstract

In this paper, the structure and working principle of four-mass vibration MEMS gyroscope (FMVMG) are introduced, and the working modes of FMVMG are simulated and analyzed. On the basis of this, an improved noise reduction method based on interval local mean decomposition (ILMD) and parabolic tracking time-frequency peak filtering (PTTFPF) is proposed. PTTFPF can resample the signal along a parabolic path and select the optimal filtering trajectory, but there is still a contradiction, choosing a short window length may lead to good signal amplitude retention, but the random noise reduction effect is not good, while choosing a long window length may lead to serious amplitude attenuation, but the random noise reduction effect is better. In order to achieve a better balance between effective signal amplitude preservation and random noise reduction, the ILMD method was used to improve PTTFPF. First, the original signal was decomposed into product functions (PFs) by local mean decomposition (LMD) method, and the sample entropy (SE) of each PF was calculated. The PFs are divided into three different components. Then, short window PTTFPF is used for useful PF and long window PTTFPF is used for mixed PF, noise PF is directly removed. Then the final signal is reconstructed. Finally, the denoised useful PF and mixed PF are reconstructed to obtain the final signal. The proposed ILMD-PTTFPF algorithm was verified by temperature experiments. The results show that the denoising performance of the ILMD-PTTFPF algorithm is better than that of traditional wavelet threshold denoising and Kalman filtering.

## 1. Introduction

With the development of sensor manufacturing technology, the accuracy of micro-electro-mechanical system (MEMS) gyroscopes has been continuously improved, which can meet the requirements of many fields. However, in the process of test and calibration of gyroscope, the noise of the test signal greatly affects the identification of the working mode of the sensor and the calibration accuracy of the sensor, therefore, reducing the noise of the output signal is an effective method to improve the calibration accuracy of the MEMS gyroscope, thereby improving the performance of the gyroscope [1,2,3].

Previously, there were many scholars devoted to the study of algorithms to suppress the noise of MEMS gyroscope. In work [4], a new type of silicon microstructure equivalent circuit model was presented, according to this model and weak signal detection technology, the different noise components of MEMS gyroscopes are studied, including mechanical thermal noise, electrical thermal noise, flicker noise, and Coriolis signal in-phase noise. Work [5] proposed an improved denoising algorithm based on fast Fourier transform (FFT) and simple wavelet denoising algorithm. Experimental results show that the performance of the improved denoising method is better than that of the FFT denoising method and simple Wavelet denoising method. Work [6] proposed a signal processing algorithm based on a three-dimensional adaptive filter demodulator. This method eliminates the common mode noise and orthogonal coupling caused by the initial capacitance mismatch. This kind of adaptive filter has the advantages of fast convergence, low noise, and less hardware resources. Work [7] proposes a drift model based on the combination of genetic algorithm and Kalman filter, which can eliminate errors in the entire temperature range. In work [8], a real-time wavelet de-noising method is used for the error compensation of MEMS gyroscope, at the same time, sparse and redundant representation methods are used to optimize the wavelet coefficients, the lag correction algorithm is used to reduce the boundary effect of wavelet decomposition. In work [9], a noise reduction algorithm based on improved empirical mode decomposition (EMD) and forward linear prediction (FLP) is proposed, which eliminates noise more effectively than traditional EMD or FLP methods. In work [10], in order to balance the signal fidelity and noise reduction effect, sample entropy (SE) and empirical mode decomposition (EMD) are used to improve time-frequency peak filtering (TFPF). The denoising performance of this method is better than traditional wavelet, Kalman Filter and fixed window length TFPF method. In work [11], a new interval empirical mode decomposition (IEMD) noise reduction method is proposed, which greatly improves the signal quality and the accuracy of the inertial navigation system solution. Work [12] proposed a filtering algorithm based on quadrature demodulation, adding Q-channel demodulation filtering to the original single-channel demodulation method. This solution can eliminate the interference signal without destroying the output signal of the gyroscope. Work [13] introduced an improved Sage-Husa adaptive Kalman filter (SHAKF), and proposed an improved autoregressive model. The SHAKF method reduces the random error of the fiber optic gyroscope effectively and improves the accuracy of the fiber optic gyroscope. Work [14] combined adaptive sampling strong tracking algorithm (ASSTA) and scaled unscented Kalman filter algorithm for interferometric fiber optic gyroscope (IFOG) signal denoising. In this algorithm, the state error covariance (P) is updated by the suboptimal fading factor and the ASSTA method. The simulation results show that the algorithm can reduce the drift of the gyroscope signal. Work [15] proposed a MEMS gyroscope denoising method based on deep learning, applying recurrent neural networks (RNN) variant simple recurrent unit (SRU-RNN) to the MEMS gyroscope’s signal denoising. The simulation results prove the SRU-RNN’s superiority. Work [16] proposed a nonlinear parabolic tracking time-frequency peak filtering (PTTFPF). This method resamples the signal along the parabolic path and extracts the data matrix to select the optimal filtering trajectory. From the literature above it can be concluded that local mean decomposition and Kalman filter are very excellent noise reduction method for MEMS gyroscopes. However, there are some other outstanding de-noising methods that have been proposed but never used for four-mass vibration MEMS gyroscope.

In recent years, time-frequency peak filtering (TFPF) has been applied more and more in the field of signal processing [17]. However, the traditional TFPF has disadvantages in the selection of window length. Using a fixed window length for all signal components will result in a serious loss of effective components or insufficient noise reduction capabilities. In addition, the PTTFPF resamples the signal along the parabolic trajectory, this method can enhance linearity and effectively improve the performance of traditional TFPF [16,17,18]. However, the curvature in the data of the FMVMG is relatively complicated, and it is difficult to accurately fit a parabola to a signal mixed with noise. In order to solve these problems, we combine PTTFPF and ILMD in this article. The interval local mean decomposition (ILMD) method is essentially a combination of local mean decomposition (LMD) [19,20,21,22] and sample entropy (SE) [23,24]. First, the traditional LMD algorithm is used to decompose the FMVMG signal into several product functions (PFs); second, based on the characteristics of sample entropy calculation and product functions, the numerous PFs are divided into three categories: noise PF, mixed PF, and useful PF. The noise PFs should be wiped off directly. The mixed PF is denoised with a long-window PTTFPF, because long-window PTTFPF can effectively reduce random noise. The useful PF is denoised with a short-window PTTFPF, this can make the linearity in the window best, so as to retain the information of the signal effectively on the basis of removing the noise. Finally, recombine the denoised mixed PF and denoised useful PF to get the final denoised signal.

The experimental results can verify the reliability of the denoising method in this paper. Compared with the source signal, the ILMD-PTTFPF method improves the bias stability of denoised signal by about 19 times, and reduces the noise characteristic by about 24 dB.

The rest of this paper is organized as follows: FMVMG description is shown in Section 2; Section 3 is a description of the proposed ILMD-PTTFPF algorithm. Section 4 introduces the experimental results and the discussion of denoising result; Section 5 is the conclusion.

## 2. Four-Mass Vibration MEMS Gyroscope

The noise reduction algorithm in this paper is aimed at a new microelectromechanical system, a four-mass vibration MEMS gyroscope (FMVMG). The core components of FMVMG are four sensitive masses, and the decoupling of the driving and detection modes of the four masses is realized through the design of a reasonable ratio of stiffness between two degrees of freedom and elastic beams. Due to the advantages of the material and structure of the resonator, FMVMG has many significant advantages, such as high precision, low energy consumption, long life, and mass production.

### 2.1. Structure and Workong Principle of FMVMG

Figure 1 shows the structure of FVMMG with four sensitive blocks at its core. The four-mass harmonic oscillator consists of 10 supporting anchors, 16 driving and detection frames, 4 mass blocks and multiple folded beams for connection, drive, and detection (the specific structure is shown in Figure 1). Four cross-shaped driving and detection frames are designed around each mass block. One end of the drive frame and the detection frame is connected with the mass block through the folding beam, and the other end is connected with the supporting anchorage point. The reasonable selection of the stiffness of the folded beam and the design of the 2-DOF structure effectively decouple the driving mode from the detection mode.

Because of the two free design, the whole four mass block is designed to connect many elastic beams. The structure is complex, so we can approximate it to a spring damping structure. Therefore, the lumped model of FMVMG is shown in Figure 2.

FMVMG is a kind of Coriolis vibration gyroscope. Its working mechanism is the mutual transfer of Coriolis force coupling energy between driving mode and detection mode. wave. Its working mode is shown in Figure 3. When there is no external input angular velocity, a periodic sinusoidal AC signal is applied to the driving electrode of each mass block to obtain the driving mode shown in Figure 3a. When the mounted carrier rotates in the direction perpendicular to the driving vibration, an angular velocity is generated on the *z*-axis, and the energy of the driving mode forms the vibration of the detection mode as shown in Figure 3b through the Coriolis effect.

### 2.2. Modal Analysis of FMVMG

The ANSYS software is used to simulate the sensitive structure of the four-mass gyro, and the simulation results of the first 12 modals are extracted. The natural frequencies are shown in Table 1. The first eight modes of the sensitive structure are translational motions in the plane of the four-mass block, in which the motions of the adjacent masses in the sixth and seventh modes are reversed, and the four-mass micro-gyroscope vibrates in a trapezoidal mode. The FMVMG studied in this paper was measured in these two modes, and the 6th and 7th modes are the basic modes of FMVMG. Through electrostatic control techniques and rational electrode arrangements, the gyroscope can resonate in these modes. The 8th to 12th modes are out-of-plane translational motions of the four-mass.

The main model simulation diagram of the four-mass resonator structure is shown in Figure 4. The natural frequencies of the resonant structure are 30,740 Hz and 30,886 Hz, respectively, and the minimum frequency difference of the interference mode is 1377 Hz.

Under the electrostatic drive, the resonator works in a driving manner, as shown in Figure 4a, the four-mass blocks move in opposite directions in the up-down direction. When measuring the angular velocity, the resonator changes the vibration mode according to the Coriolis principle, and excites the detection mode as shown in Figure 4b, and the four-mass block moves in the opposite direction in the left and right directions.

## 3. Denoising Methods

### 3.1. Interval Local Mean Decomposition

Time-frequency analysis is an effective tool for analyzing non-stationary signals. The time-frequency analysis method provides joint distribution information of time domain and frequency domain. It describes the relationship between the frequency of a signal and time. The basic principle of time-frequency analysis is to design a joint function of time and frequency, and use it to describe the energy intensity of a signal at different times and frequencies. Typical time-frequency analysis methods include window Fourier transform, Wigner distribution, wavelet transform, etc., [25,26].

Local mean decomposition (LMD) is also a time–frequency analysis method. It can decompose non-stationary signals into numerous product functions (PF) adaptively. PF is the product of the frequency modulation signal and the envelope signal. The amplitude of PF represents the instantaneous amplitude of the envelope signal, and its instantaneous frequency is obtained from the frequency modulation signal. Therefore, the parameters of each PF component can represent the complete time-frequency distribution of the original signal.

The process of LMD is shown as follows [19]:Step 1: determine all local extreme points *n_i_* of the original signal *x*(*t*), and calculate the mean value *m_i_* of two adjacent extreme points *n_i_* and *n_i+1_*:
(1)$$m_{i}=\frac{\left(n_{i}+n_{i+1}\right)}{2}$$

All *m_i_* is connected by straight lines and smoothed by moving average method to obtain the local mean function *m_11_*(*t*).

Step 2: envelope estimates *a_i_* are calculated from local mean points *n_i_*:


(2)
$$a_{i}=\frac{\left|n_{i}-n_{i+1}\right|}{2}$$


Similarly, all the *a_i_* are connected by straight lines and smoothed by moving average method to obtain the envelope estimation function *a_11_*(*t*).

Step 3: the local mean function *m_11_*(*t*) is separated from the original signal *x*(*t*):


(3)
$$h_{11}(t)=x(t)-m_{11}(t)$$


Step 4: divide *h_11_*(*t*) by *a_11_*(*t*) to demodulate *h_11_*(*t*):


(4)
$$s_{11}(t)=\frac{h_{11}(t)}{a_{11}(t)}$$


Ideally, *s_11_*(*t*) is a pure frequency-modulated signal, in which case, the envelope estimation function *a_12_*(*t*) = 1. If *s_11_*(*t*) does not meet the conditions, then *s_11_*(*t*) will be used as the original data to repeat the above iterative process until a pure frequency-modulated signal *s_1n_*(*t*) is obtained, that is, −1≤ *s_1n_*(*t*) ≤1, and its envelope estimation function *a_1(n+1)_*(*t*) = 1.
(5)$$\left\{\begin{array}{c} h_{11}(t)=x(t)-m_{11}(t)\\ h_{12}(t)=s_{11}(t)-m_{12}(t)\\ \vdots \\ h_{1n}(t)=s_{1(n-1)}(t)-m_{1n}(t) \end{array}\right.$$

Among them:(6)$$\left\{\begin{array}{c} s_{11}(t)=h_{11}(t)/a_{11}(t)\\ s_{12}(t)=h_{12}(t)/a_{12}(t)\\ \vdots \\ s_{1n}(t)=h_{1n}(t)/a_{1n}(t) \end{array}\right.$$

The condition for iteration termination is:(7)$$\underset{n\to \infty }{\lim}a_{1n}(t)=1$$

Step 5: the envelope signal is obtained by multiplying all the envelope estimation functions generated in the iteration process:


(8)
$$a_{1}(t)=a_{11}(t)\cdot a_{12}(t)\cdots a_{1n}(t)=\prod_{i=1}^{n}a_{1i}(t)$$


Step 6: the first PF component *PF_1_*(*t*) of the original signal is obtained by multiplying the envelope signal *a_1_*(*t*) and the pure frequency-modulated signal *s_1n_*(*t*):


(9)
$$PF_{1}(t)=a_{1}(t)\cdot s_{1n}(t)$$


*PF_1_*(*t*) contains the component with the highest frequency in the original signal, which is a single-component AM-FM signal. Its instantaneous amplitude is the envelope signal *a_1_*(*t*), and its instantaneous frequency *f_1_*(*t*) can be calculated from the pure frequency-modulated signal *s_1n_*(*t*):(10)$$f_{1}(t)=\frac{1}{2\pi }\cdot \frac{d[\arccos(s_{1n}(t))]}{dt}$$

Step 7: the first PF component *PF_1_*(*t*) is separated from the original signal to obtain a new signal *u_1_*(*t*), using *u_1_*(*t*) as the original data, repeat the above steps for *k* times until *u_1_*(*t*) is a monotone function.


(11)
$$\left\{\begin{array}{c} u_{1}(t)=x(t)-PF_{1}(t)\\ u_{2}(t)=u_{1}(t)-PF_{2}(t)\\ \vdots \\ u_{k}(t)=u_{k-1}(t)-PF_{k}(t) \end{array}\right.$$


At this time, the original signal *x*(*t*) is decomposed into the sum of PF components and a monotone function *u_k_*(*t*):(12)$$x(t)=\sum_{i=1}^{k}PF_{i}(t)+u_{k}(t)$$

Since the result of LMD contains many PF components, it is impossible to denoise all PF components separately. Therefore, for the convenience of subsequent noise reduction processing, sample entropy (SE) is introduced here to classify numerous PF components, which can be divided into noise PF, mixed PF, and useful PF. The interval local mean decomposition (ILMD) method is essentially a combination of LMD and SE.

SE is used to judge the complexity of time series by measuring the probability of generating new patterns. For the time series composed of N data, the calculation method of sample entropy is as follows:Step 1: construct a sequence of m-dimensional vectors by ordinal number, *X_m_*(*1*), *X_m_*(*2*), ⋯, *X_m_*(*N-m + 1*), among them, *X_m_*(*i*) = { *x*(*i*), *x*(*i + 1*), …, *x*(*i + m − 1*)}, 1≤ *i* ≤*N-m+1*.Step 2: the distance *d*[*X_m_*(*i*), *X_m_*(*j*)] between vectors *X_m_*(*i*) and *X_m_*(*j*) is defined as the absolute value of the maximum difference between the two corresponding elements:
(13)$$d[X_{m}(i),X_{m}(j)]=\max_{k=0,\cdot \cdot \cdot ,m-1}(\left|x(i+k)-x(j+k)\right|)$$

Step 3: count the number of *j* (1≤ *j* ≤*N-m*, *j*≠*i*) whose distance between *X_m_*(*i*) and *X_m_*(*j*) is less than *r*, and define it as *B_i_*:


(14)
$$B_{i}^{m}(r)=\frac{1}{N-m+1}B_{i}$$


where, 1≤ *i* ≤*N-m*+1.

Step 4: definition *B^(m)^*(*r*) is:


(15)
$$B_{}^{\left(m\right)}(r)=\frac{1}{N-m}\sum_{i=1}^{N-m}B_{i}^{m}(r)$$


Step 5: increase the dimension to *m+1*, count the number of *j* (1 ≤ *j* ≤ *N-m*, *j*≠*i*) whose distance between *X_m+1_*(*i*) and *X_m+1_*(*j*) is less than *r*, and define it as *A_i_*:


(16)
$$A_{i}^{m}(r)=\frac{1}{N-m-1}A_{i}$$


Step 6: definition *A^(m)^*(*r*) is:


(17)
$$A_{}^{\left(m\right)}(r)=\frac{1}{N-m}\sum_{i=1}^{N-m}A_{i}^{m}(r)$$


*B^(m)^*(*r*) is the probability of two sequences matching *m* points under the similar tolerance *r*, *A^(m)^*(*r*) is the probability of two sequences matching *m + 1* points, and the sample entropy is defined as:(18)$$SampEn(m,r,N)=-\ln[\frac{A^{\left(m\right)}(r)}{B^{\left(m\right)}(r)}]$$
where *N* is the length of the signal; *m* is the embedded dimension, it usually takes a value of 1 or 2; *r* is the similar tolerance (usually selecting 10–25% of the signal standard deviation).

### 3.2. Parabolic Tracking Time Frequency Peak Filtering

Time-frequency peak filtering (TFPF) realizes signal denoising by encoding the noisy signal *x*(*t*) as the instantaneous frequency (IF) of an analytic signal *z*(*t*):(19)$$z(t)=a(t)\cdot e^{j2\pi \mu \int_{0}^{t}x(\lambda )d\lambda }$$
where, *μ* is the frequency modulation (FM) index, *a*(*t*) is the instantaneous amplitude of *z*(*t*).

Pseudo Wigner–Ville distribution (PWVD)is a bilinear time-frequency distribution with good time–frequency focusing. It is an effective method for suppressing cross terms in Wiener–Ville distribution (WVD). The denoised signal *x’*(*t*) can be recovered by using the peak of the PWVD of *Z*(*t*):(20)$$x\prime (t)=\frac{\arg\max[PW_{z}(t,f)]}{\mu }$$
where,
(21)$$PW_{z}(t,f)=\int_{-\infty }^{\infty }h(\tau )z(t+\frac{\tau }{2})(t-\frac{\tau }{2})e^{j2\pi f\tau }d\tau$$
where *h*(*τ*) is the window function.

The window length (WL) in TFPF directly affects the signal fidelity and noise reduction effect. The long-WL TFPF has good denoising performance, but the denoised signal will have amplitude loss, especially in the peak and trough position. TFPF with short-WL has good linearity in the window, and the signal amplitude loss is small after denoising, but there are still many noise components.

Parabolic-tracking time–frequency peak filtering (PTTFPF) is an improved method based on TFPF. This method can improve the performance of traditional TFPF by using the selection of filtering trajectory and data resampling.

Canny edge detection (CED) algorithm can be used to select the optimal filtering trace [27,28]. The process of CED includes Gaussian low-pass filtering, calculation of gradient value and direction, non-maximum suppression of gradient value, and edge connection.

Data resampling is the process of transforming the signal to a new data matrix along the parabolic filter trace. After the resampling, the linearity of the signal will be greatly improved, so as to improve the noise reduction performance of TFPF in the window.

### 3.3. Step of ILMD-PTTFPF

In order to further improve the noise reduction performance of PTTFPF, ILMD and PTTFPF are combined in this paper. The specific steps are as follows:Step 1: Local mean decomposition

Through local mean decomposition of MSRVG output signals, PFs with different instantaneous frequencies can be obtained.

Step 2: Classify the PF by SE value

Since the results of LMD contain many PF components, it is not possible to denoise all PF components separately. In order to simplify the subsequent noise reduction processing, the PFs are classified by sample entropy, which can be divided into noisy PF, mixed PF, and useful PF.

Step 3: Denoise for different PF

Select PTTFPF with different WL to denoise mixed PF and useful PF, and eliminate noisy PF.

Step 4: Signal reconstruction

The final denoised signal can be obtained by adding denoised mixed PF and useful PF. The steps of ILMD-PTTFPF method are shown in Figure 5.

## 4. Experiment

### 4.1. Experiment Platform and Equipment

We carry out temperature experiment to obtain the output signal of FMVMG in different temperature environment. The main equipment used in temperature experiment are temperature control box, data acquisition card, signal generator, and the prototype of FMVMG. The experimental equipment is shown in Figure 6 and Figure 7. The temperature control box in the laboratory can control the temperature accurately, we have carried out the experiment in the temperature range of −40 °C to 60 °C. The computer beside the device is used to collect the output signal of the FMVMG.

The prototype of FMVMG is shown in Figure 8. First, put the gyroscope into the temperature control box, reduce the temperature to −40 °C, and keep it for one hour. After the gyroscope runs stably at this temperature, set the temperature change range from −40 °C to 60 °C. Keep it for about 1.5 h after the temperature rises by 20 °C to ensure the stability of the internal temperature change of the gyroscope. During this period, the data acquisition is continuous, and the sampling rate is 1 Hz, the test lasted 32,290 s.

In order to avoid contingency, we conducted five groups of temperature experiments, and found that the results were generally consistent. One group was selected as the sample data for analysis and processing. The collected data are the output voltage, which can be converted to angular velocity by scaling.

### 4.2. Denoising Processing

According to the ILMD-PTTFPF algorithm, the first step is the local mean decomposition of the original signal. The output signal of FMVMG is decomposed into eight PFs with different physical properties and one residual component. The product function is obtained by multiplying the envelope signal with the frequency-modulated signal. The complete time-frequency distribution of the original signal can be obtained by combining the parameters of each PF. Figure 8 is the result of LMD, where *X*(*t*) is the original output signal of FMVMG and *U*(*t*) is the residual component.

According to the results of the LMD in Figure 9, if all the PFs are processed by filters, we need to study eight filters. Obviously, the work is too complex if eight filters are employed. Therefore, it is necessary to simplify the filtration process. ILMD is to simplify the results of LMD into three PFs by SE values. Therefore, the second step is to calculate the sample entropy value of each PF to distinguish the complexity of them, and classify PFs according to the similarity of SE value. The sample entropy interval should be divided into three parts evenly. The PFs in the minimum interval is considered as useful PF, while the PFs in the middle interval is considered as mixed PF, and the noise PF is in the maximum interval. The ideal useful PF is a pure signal without noise, the noise PF should be pure noise, and the mixed PF contains pure signal and noise.

Figure 10 shows the calculated results of sample entropy value and PF classification results. According to the calculation results, SE values in the range of 0~0.55 are considered to be useful PF (PF4, PF5, PF6, PF7, PF8), PTTFPF with short window length should be used to denoise the useful PF, so as to retain the information of the signal. SE values in the range of 0.55~1.1 are considered mixed PF (PF2 and PF3), using long window PTTFPF can obtain the best noise reduction performance. Noise PF should be directly removed, which is in the range of 1.1~1.67 (PF1).

Figure 11 shows the denoising results of noise PF, useful PF, and mixed PF respectively. It can be seen that after the long-WL PTTFPF denoising, the noise of mixed PF is significantly reduced, and the original signal features are not lost at the peak signal. Useful PF also retains the information of the original signal well after short-WL PTTFPF denoising.

By superimposing the denoised useful PF and the denoised mixed PF, we obtain the final denoised signal.

### 4.3. Discussion of the Denoising Results

Figure 12 shows the denoising results using ILMD-PTTFPF method and the comparison between different denoising algorithms. It can be seen that although traditional wavelet threshold denoising and Kalman filtering can reduce part of the signal noise, it is obvious that the ILMD-PTTFPF method in this paper has the best denoising ability.

Indicators of signals after denoising by each method are shown in Table 2 [7,22]. Allan variance is a standard gyroscope performance analysis method recognized by IEEE and it is widely used [29]. Comparison of Allan variance of three noise reduction methods is shown in Figure 12. The bias stability can reflect the random drift characteristic of the gyroscope. The bias stability is an important indicator to reflect the performance of the gyroscope among the performance indicators of the gyroscope, the smaller the bias stability of the gyroscope, the stronger the performance of the gyroscope. The signal to noise ratio (SNR) is the ratio of the effective power of the useful signal to the noise signal, the higher the SNR value, the lower the noise of the signal. Standard deviation can reflect the degree of dispersion of a data set, and it can also indicate the strength of signal noise.

It can be seen from Table 2 that compared with the original signal, the bias stability of signal denoised by ILMD-PTTFPF method is improved by about 19 times, it is optimized by 94.8%. The SNR is improved by about 24 dB (noise characteristics are reduced by 24 dB). The comparison results in Figure 13 show that the ILMD-PTTFPF method has better noise reduction performance than traditional wavelet threshold denoising and Kalman filtering.

## 5. Conclusions and Discussion

This paper introduces the structure and working principle of four-mass vibration MEMS gyroscope (FMVMG). On this basis, a noise reduction algorithm based on interval local mean decomposition (ILMD) and parabolic tracking time-frequency peak filtering (PTTFPF) is proposed. The IEMD-TFPF method in this paper is different from the previous noise reduction methods. First, ILMD is introduced to process the output signal of the FMVMG, and the original signal is divided into useful signal, mixed signal, and noise signal. Then, PTTFPF with different window length is used to denoise the three kinds of signals respectively, so as to improve the performance of PTTFPF. Finally, the effectiveness of the algorithm is verified by temperature experiment. Experimental and comparative results show that compared with the original signal, the bias stability of signal denoised by ILMD-PTTFPF method is improved by about 19 times, it is optimized by 94.8%, and the SNR is improved by about 24 dB (noise characteristics are reduced by 24 dB). However, the bias stability of signal denoised by wavelet threshold denoising method is improved by about 1 time, it is optimized by 30.4%, and the SNR is improved by about 12.95 dB (noise characteristics are reduced by 12.95 dB); the bias stability of signal de-noised by Kalman filter method is improved by about four times, it is optimized by 75.7%, and the SNR is improved by about 17.41 dB (noise characteristics are reduced by 17.41 dB). Compared with wavelet threshold denoising and Kalman filtering, the proposed ILMD-PTTFPF method has the best denoising ability, and ILMD-PTTFPF method can effectively suppress the noise of the FMVMG.

## Figures and Tables

**Figure 1 micromachines-13-01807-f001:**
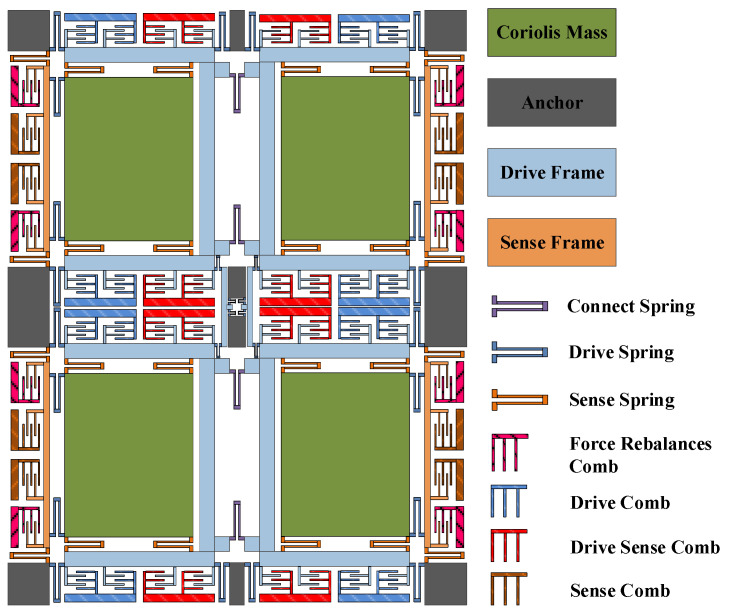
Structure of FMVMG.

**Figure 2 micromachines-13-01807-f002:**
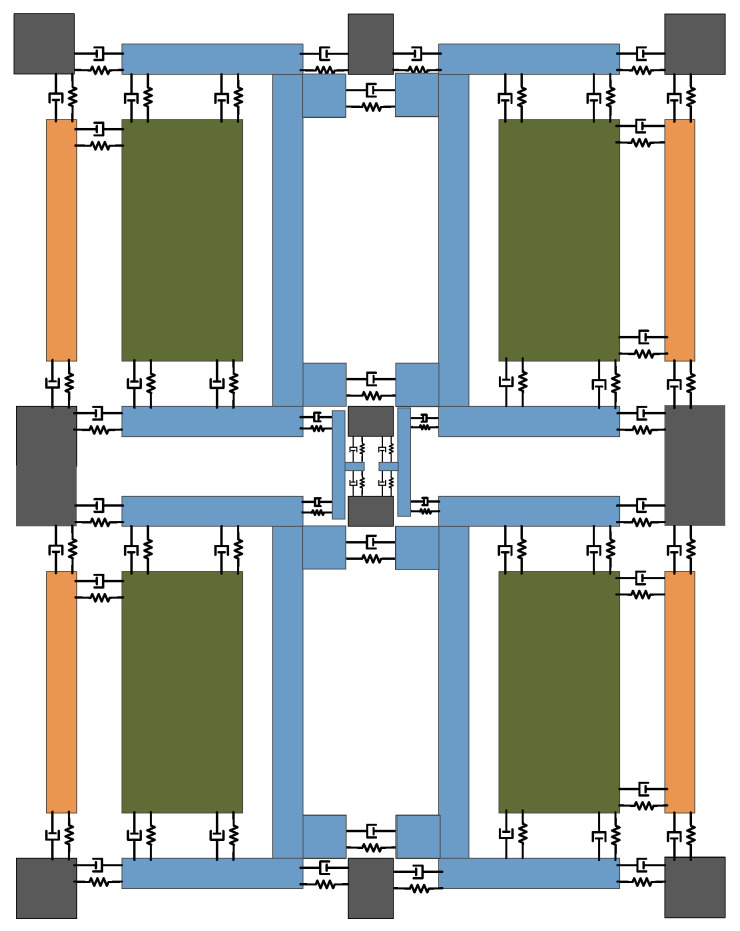
The lumped model of FMVMG.

**Figure 3 micromachines-13-01807-f003:**
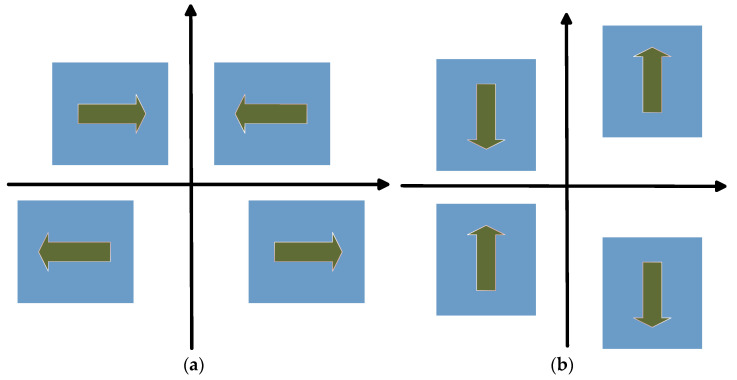
The working mode of FMVMG. (**a**) Drive model; (**b**) Sense model.

**Figure 4 micromachines-13-01807-f004:**
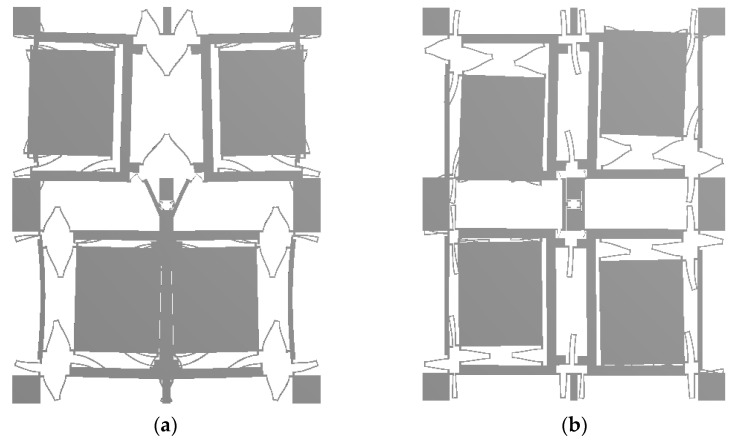
Working mode simulation diagram of FMVMG. (**a**) Drive model; (**b**) Sense model.

**Figure 5 micromachines-13-01807-f005:**
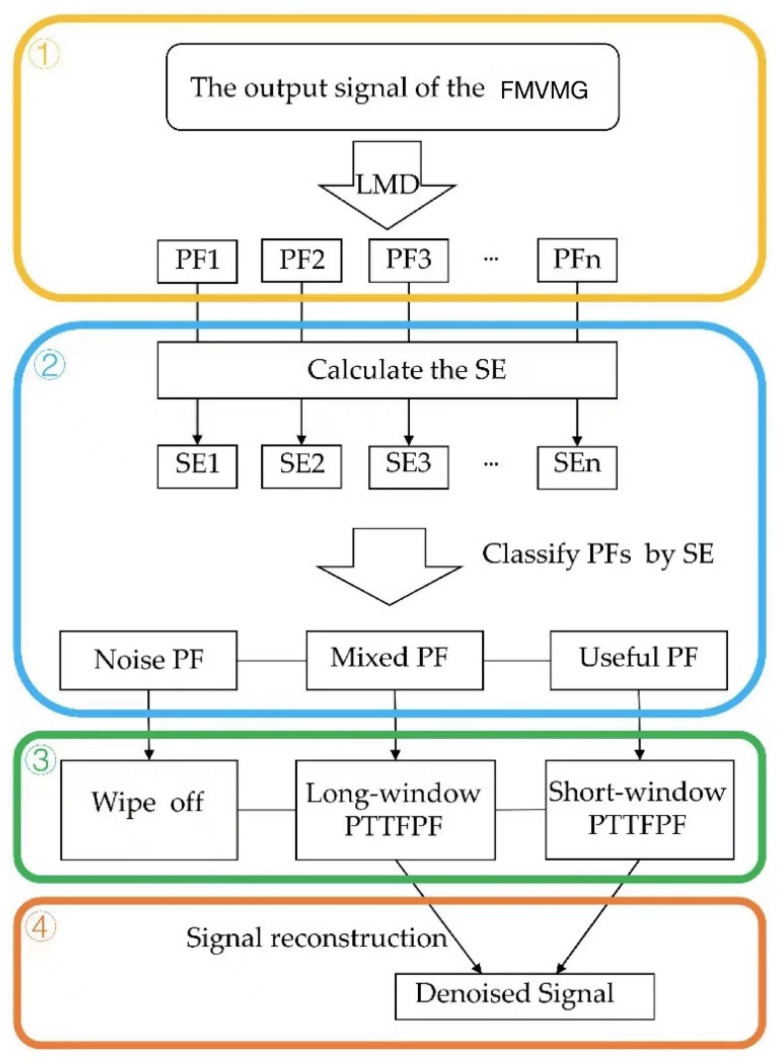
The steps of ILMD-PTTFPF method.

**Figure 6 micromachines-13-01807-f006:**
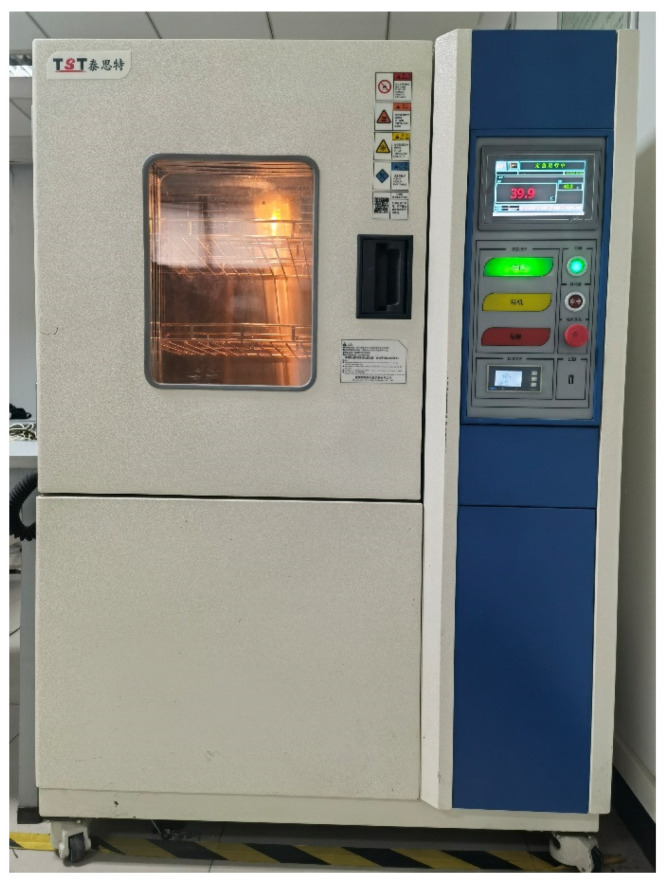
Experimental apparatus for FMVMG.

**Figure 7 micromachines-13-01807-f007:**
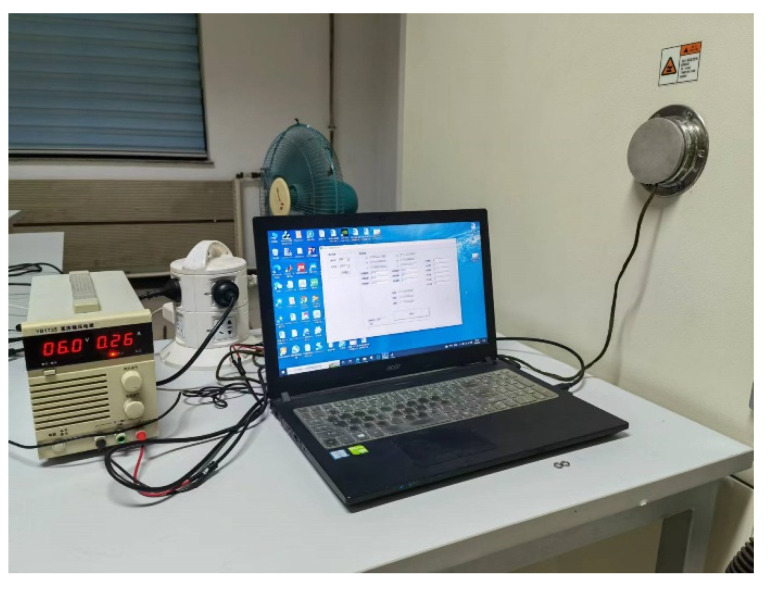
Experimental apparatus for FMVMG.

**Figure 8 micromachines-13-01807-f008:**
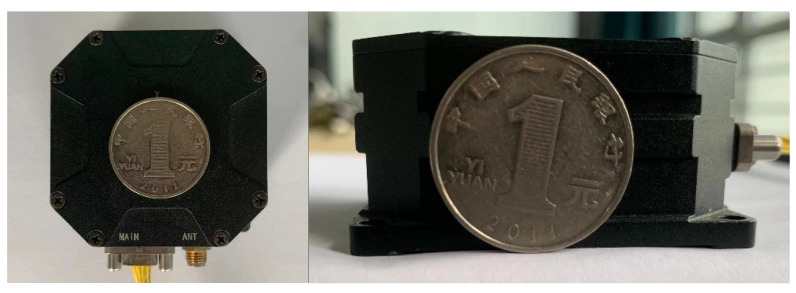
The prototype of FMVMG.

**Figure 9 micromachines-13-01807-f009:**
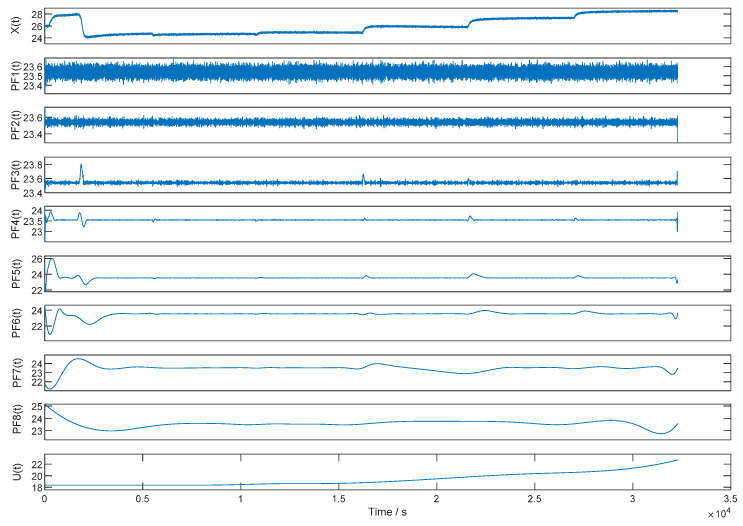
The results of LMD.

**Figure 10 micromachines-13-01807-f010:**
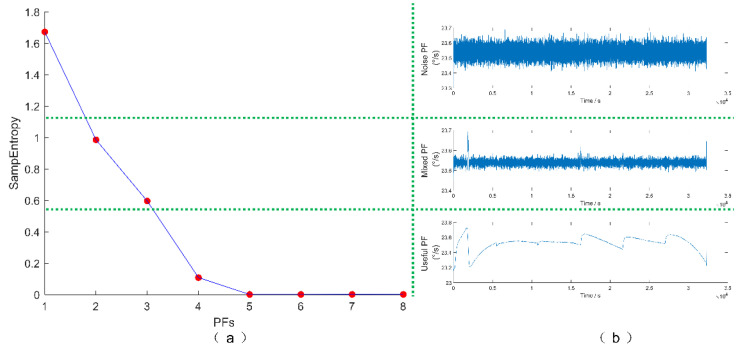
The calculated results of SE value and PF classification results.(**a**) Sample entropy; (**b**) PF classification.

**Figure 11 micromachines-13-01807-f011:**
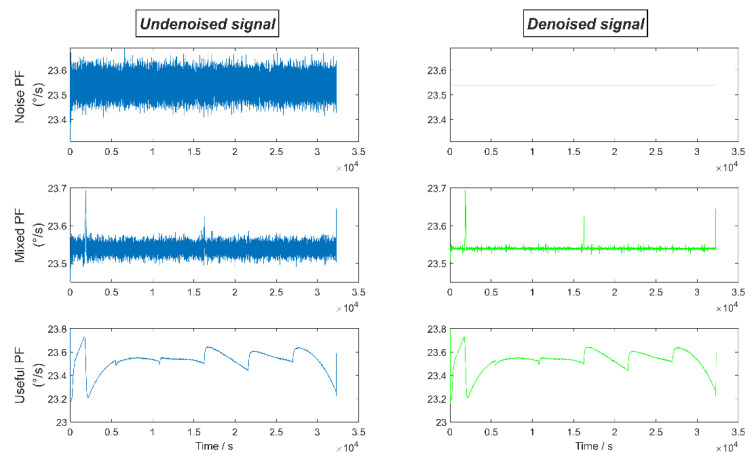
The denoising results of noise PF, useful PF, and mixed PF, respectively.

**Figure 12 micromachines-13-01807-f012:**
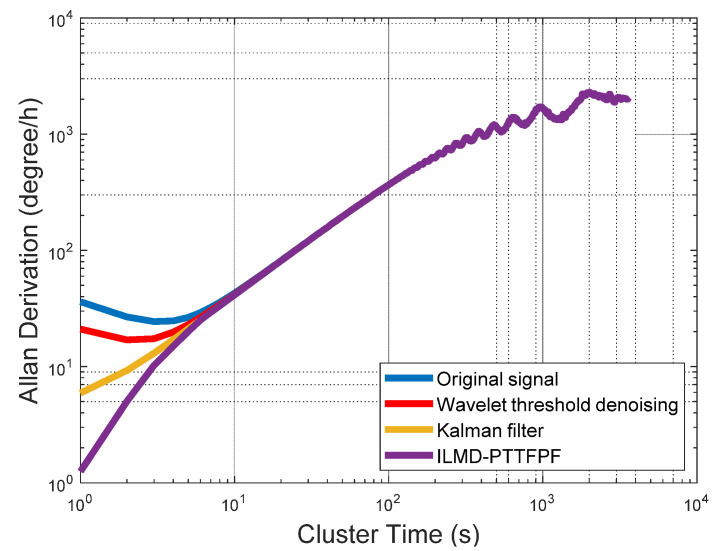
Allan variance of different noise reduction methods.

**Figure 13 micromachines-13-01807-f013:**
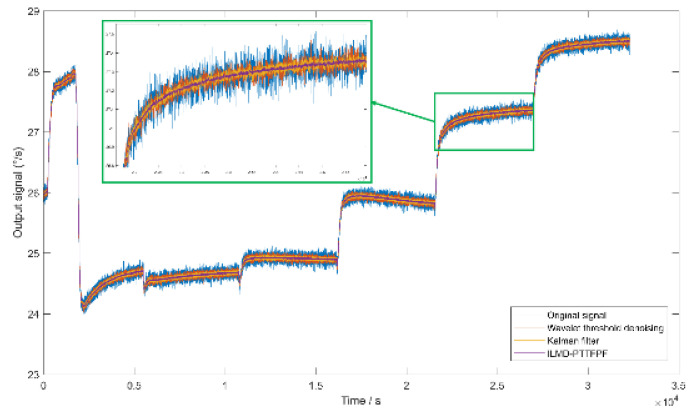
Comparison of different noise reduction methods.

**Table 1 micromachines-13-01807-t001:** The oscillator modal frequency of FMVMG.

Modal Order Number	Resonant Frequencies
1	25,644 Hz
2	25,999 Hz
3	26,476 Hz
4	26,550 Hz
5	29,636 Hz
6	30,740 Hz
7	30,886 Hz
8	37,230 Hz
9	38,543 Hz
10	40,070 Hz
11	41,445 Hz
12	41,955 Hz

**Table 2 micromachines-13-01807-t002:** Comparison of indexes of different noise reduction methods.

Methods	Bias Stability (°/h)	Standard Deviation	Signal to Noise Ratio (dB)	References
Original signal	24.44	0.0108	22.5	-
Wavelet threshold denoising	16.99	0.00567	35.45	Cao et al. [22]
Kalman filter	5.93	0.00734	39.91	Cao et al. [7]
ILMD-PTTFPF	1.26	0.00455	46.16	-

## Data Availability

Not applicable.

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
