# Peer review of "A Noise Reduction Method for Four-Mass Vibration MEMS Gyroscope Based on ILMD and PTTFPF"

_micromachines, 2022, doi:10.3390/mi13111807_

Round 1

Reviewer 1 Report

In the article titled “A Noise Reduction Method for Four-Mass Vibration MEMS Gyroscope Based on ILMD and PTTFPF” propose a concept to noise reduction based on interval local mean decomposition (ILMD) and parabolic tracking time-frequency peak filtering (PTTFPF). for a better understanding it should be integrated considering the following aspects:

Overall, I do not find this work exciting due to following reasons

1.There are many simple approaches for noise reduction. Authors didn't compare their results and novelty even with the other methods.

2. The authors didn’t provide a great significance of work in their introduction part.

Author Response

Dear Reviewer:

Thank you so much for professional suggestions, followed which we improved a lot. 

Bests,

Huiliang Cao

Reviewer 2 Report

Dear Authors,

In this manuscript, the Authors presented a concise algorithm a noise reduction method for the four-mass vibration MEMS gyroscope. For this purpose, they combined the interval local mean decomposition and the parabolic tracking time-frequency peak filtering. Thus, they could demonstrate that thanks to this combination of such methods, ie ILMD and PTT-FPF, the tested denoising performance is better than in the case of using the more popular wavelet threshold denosing and Kalman filtering algorithms. It should be emphasized here that the algorithm has been discussed in detail and, additionally, its subsequent steps have been presented graphically. Then, the Authors conducted an experiment at various temperatures (from -40 to +60 degrees Celsius) in order to obtain the original ouput signal of FMVMG, necessary for local decomposition in the denoising process. The obtained results of denoising of noise PF, useful PF and mixed PF were plotted (Figure 10) and analyzed in detail (Fig. 11 and Fig. 12). In the summary, the authors emphasized that the comparison of the results obtained from the proposed algorithm with the original signal showed that bias stability and signal denoised were improved about 19 times, i.e. noise characteristics were reduced by 24 dB. Similarly, the comparison of these two methods (proposed and classic) shows that the new method (or rather the combination of ILMD with PTT-FPF) is more effective in terms of denoising.

Unfortunately, a big disadvantage of the manual (according to the reviewer) is a very cursory description of the four-mass vibration structure of the MEMS gyroscope. It is still not very popular and at the same time very complex type of gyroscope, so the principle of its operation would be more understandable with a detailed description of the structure. The following notes apply to Figure 1, where there are no specifications of the main parts of Four-Mass Vibration Gyroscope MEMS (FMVMG) - preferably using cross-references. For example, where are:

1) masses: driving, decoupled, proof and detection,

2) anchors

3) springs: drive, connect, sense

4) comb: drive, sense, drive sense,

5) force rebulances comb, etc.

In addition there is no lumped model of FMVMG, sufficient to verify the eigenfrequencies calculated with the use of FEM.

Yours sincerely, Reviewer.

Author Response

Dear Reviewer:

Thank you so much for your professinoal suggestions, we  appreciate that very much. And the paper improved a lot.

Bests,

Huiliang Cao

Round 2

Reviewer 1 Report

Most of the questions have answered based on the previous comments. However, there is no references in the Table 2 when comparing to other methods. Please add some references!

Author Response

Dear Expert, we have revised the paper as you suggested, thank you so much for your suggestion!